# Semi-Discrete Normalizing Flows through Differentiable Tessellation

**Ricky T. Q. Chen**
Meta AI
rtqichen@meta.com

**Brandon Amos**
Meta AI
bamos@meta.com

**Maximilian Nickel**
Meta AI
maxn@meta.com

## Abstract

Mapping between discrete and continuous distributions is a difficult task and many have had to resort to heuristical approaches. We propose a tessellation-based approach that directly learns quantization boundaries in a continuous space, complete with exact likelihood evaluations. This is done through constructing normalizing flows on convex polytopes parameterized using a simple homeomorphism with an efficient log determinant Jacobian. We explore this approach in two application settings, mapping from discrete to continuous and vice versa. Firstly, a *Voronoi dequantization* allows automatically learning quantization boundaries in a multidimensional space. The location of boundaries and distances between regions can encode useful structural relations between the quantized discrete values. Secondly, a *Voronoi mixture model* has near-constant computation cost for likelihood evaluation regardless of the number of mixture components. Empirically, we show improvements over existing methods across a range of structured data modalities.

## 1 Introduction

Likelihood-based models have seen increasing usage across multiple data modalities. Across a variety of modeling approaches, the family of normalizing flows stands out as a large amount of structure can be incorporated into the model, aiding its usage in modeling a wide variety of domains such as images [7, 23], graphs [29], invariant distributions [2, 25] and molecular structures [42]. However, the majority of works focus on only continuous functions and continuous random variables. This restriction can make it difficult to apply such models to distributions with implicit discrete structures, *e.g.* distributions with discrete symmetries, multimodal distributions, distributions with holes.

In this work, we incorporate discrete structure into standard normalizing flows, while being entirely composable with any other normalizing flow. Specifically, we propose a homeomorphism between the unbounded domain and convex polytopes, which are defined through a learnable tessellation of the domain, *i.e.* a set of disjoint subsets that together fully cover the domain. This homeomorphism is cheap to compute, has a cheap inverse, and results in an efficient formulation of the resulting change in density. In other words, this transformation is highly scalable—at least computationally.

Our method has the potential to be useful for a variety of applications. Firstly, the learned tessellation naturally allows defining a dequantization method for discrete variables. This allows likelihood-based models that normally only act on continuous spaces, such as normalizing flows, to work directly on discrete data in a flexible choice of embedding space with the ability to capture relational structure. Secondly, if we take each convex polytope as the support of a single mixture component, then the full tessellation defines a mixture model with disjoint components. This allows us to scale to mixtures of normalizing flows while retaining the compute cost of a single model. Following semi-discrete optimal transportation [37], which defines couplings between discrete and continuous measures, we refer to our models as semi-discrete normalizing flows that learn transformations between discrete and continuous random variables.

36th Conference on Neural Information Processing Systems (NeurIPS 2022).

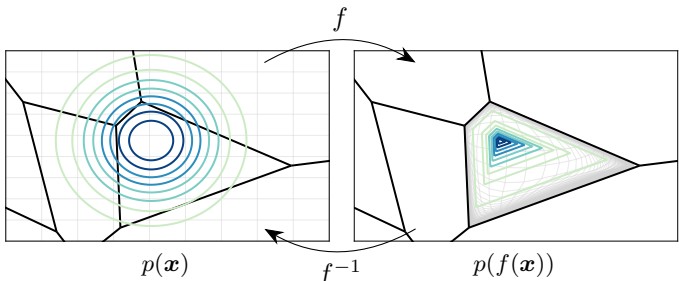

$$f$$

$$p(\boldsymbol{x}) \qquad f^{-1} \qquad p(f(\boldsymbol{x}))$$

Figure 1: We propose an invertible mapping $f$ between $\mathbb{R}^D$ and a convex polytope, which is parameterized based on a differentiable Voronoi tessellation of $\mathbb{R}^D$. This mapping adds discrete structure into normalizing flows, and its inverse $f^{-1}$ and log determinant Jacobian can both be efficiently computed.

## 2  Preliminaries

**Normalizing Flows**  This family of generative models typically includes any model that makes use of invertible transformations $f$ to map samples between distributions. The relationship between the original and transformed density functions have a closed form expression,

$$p_z(f(\boldsymbol{x})) = p_x(\boldsymbol{x}) \left| \det \tfrac{\partial f(\boldsymbol{x})}{\partial \boldsymbol{x}} \right|^{-1}. \qquad (1)$$

For instance, $p_z$ can be a simple base distribution while $f$ is learned so that the resulting $p_x$ is close to some target distribution. Many different choices of $f$ have been discussed in the literature, leading to different trade-offs and use cases [24, 36].

Furthermore, if the domain and codomain of $f$ are different, then $p_x$ and $p_z$ can have different supports, *i.e.* regions where probability is non-zero. Currently, existing designs of $f$ that have this property act independently for each dimension, such as the logit transform [7]. To the best of our knowledge, the use of general support-modifying invertible transformations have not been discussed extensively in the literature.

**Dequantization**  In order to model discrete data with density models, a standard approach is to combined with dequantization methods. These methods provide a correspondence between discrete values and convex subsets, where a discrete variable $y \in \mathcal{Y}$ is randomly placed within a disjoint subset of $\mathbb{R}^D$. There is also a correspondence between the likelihood values of the discrete random variable and the dequantized random variable [44].

Let $A_y$ denote the subset corresponding to $y$ and let $q(\boldsymbol{x}|y)$ be the dequantization model which has a bounded support in $A_y$. Then any density model $p(\boldsymbol{x})$ with support over $\mathbb{R}^D$ satisfies

$$\log p(y) \geq \mathbb{E}_{\boldsymbol{x} \sim q(\boldsymbol{x}|y)} \left[ \log(\mathbb{1}_{[\boldsymbol{x} \in A_y]} p(\boldsymbol{x})) - \log q(\boldsymbol{x}|y) \right] = \mathbb{E}_{\boldsymbol{x} \sim q(\boldsymbol{x}|y)} \left[ \log p(\boldsymbol{x}) - \log q(\boldsymbol{x}|y) \right] \quad (2)$$

Thus with an appropriate choice of dequantization, maximizing the likelihood under the density model $p(\boldsymbol{x})$ is equivalent to maximizing the likelihood under the discrete model $p(y)$.

In existing dequantization methods, the value of $D$ and the choice of subsets $A_y$ are entirely dependent on the type of discrete data (ordinal vs non-ordinal) and the number of discrete values $|\mathcal{Y}|$ in the non-ordinal case. Furthermore, the subsets $A_y$ do not interact with one another and are fixed during training. In contrast, we conjecture that important relations between discrete values should be modeled as part of the parameterization of $A_y$, and that it'd be useful to be able to automatically learn the boundaries of $A_y$ based on gradients.

**Disjoint mixture models**  Building mixture models is one of the simplest methods for creating more flexible distributions from simpler one, and mixture models can typically be shown to be universal density estimators [12]. However, in practice this often requires a large number of mixture components, which quickly becomes computationally expensive. This is because evaluating the likelihood under a mixture model requires evaluating the likelihood of each component. To alleviate this issue, Dinh et al. [8] recently proposed to use components with disjoint support. In particular, let $\{A_k\}_{k=1}^K$ be disjoint subsets of $\mathbb{R}^D$, such that each mixture component is defined on one subset and

has support restricted to that particular subset. The likelihood of the mixture model then simplifies to

$$p(\boldsymbol{x}) = \sum_{k=1}^{K} p(\boldsymbol{x}|k)p(k) = \sum_{k=1}^{K} \mathbb{1}_{[\boldsymbol{x} \in A_k]} p(\boldsymbol{x}|k)p(k) = p(\boldsymbol{x}|k=g(\boldsymbol{x}))p(k=g(\boldsymbol{x})) \quad (3)$$

where $g : \mathbb{R}^D \to \{1, \dots, K\}$ is a set identification function that satisfies $\boldsymbol{x} \in A_{g(\boldsymbol{x})}$. This framework allows building large mixture models while no longer having a compute cost that scales with $K$. In contrast to variational approaches with discrete latent variables, the use of disjoint subsets provides an exact (log-)likelihood and not a lower bound.

## 3  Voronoi Tessellation for Normalizing Flows

We first discuss how we parameterize each subset as a convex polytope, and in the next part, discuss constructing distributions on each convex polytope using a differentiable homeomorphism.

**Parameterizing disjoint subsets**   We separate the domain into subsets through a Voronoi tessellation [48]. This induces a correspondence between each subset and a corresponding *anchor point*, which provides a differentiable parameterization of the tessellation for gradient-based optimization.

Let $X = \{\boldsymbol{x}_1, \dots, \boldsymbol{x}_K\}$ be a set of anchor points in $\mathbb{R}^D$. The *Voronoi cell* for each anchor point is

$$V_k \triangleq \{\boldsymbol{x} \in \mathbb{R}^D : \|\boldsymbol{x_k} - \boldsymbol{x}\| < \|\boldsymbol{x}_i - \boldsymbol{x}\|, i = 1, \dots, K\}, \quad (4)$$

*i.e.* it defines a subset containing all points which have the anchor point as their nearest neighbor. Together, these subsets $\{V_k\}_{k=1}^{K}$ form a *Voronoi tessellation* of $\mathbb{R}^D$. Each subset can equivalently be expressed in the form of a convex polytope,

$$V_k = \{\boldsymbol{x} \in \mathbb{R}^D : \boldsymbol{a}_i^\mathsf{T} \boldsymbol{x} < b_i \; \forall i \neq k\}, \quad \text{where } \boldsymbol{a}_i = 2(\boldsymbol{x}_i - \boldsymbol{x}_k)^\mathsf{T}, b_i = \|\boldsymbol{x}_i\|^2 - \|\boldsymbol{x}_k\|^2. \quad (5)$$

For simplicity, we also include box constraints so that all Voronoi cells are bounded in all directions.

$$V_k = \{\boldsymbol{x} \in \mathbb{R}^D : \underbrace{\boldsymbol{a}_i^\mathsf{T} \boldsymbol{x} < b_i \; \forall i \neq k}_{\text{Voronoi cell}}, \quad \underbrace{\mathbf{c}^{(l)} < \boldsymbol{x} < \mathbf{c}^{(r)}}_{\text{box constraints}}\} \quad (6)$$

Thus, the learnable parameters of this Voronoi tessellation are the anchor points $\{\boldsymbol{x}_1, \dots, \boldsymbol{x}_K\}$, and the box constraints $\mathbf{c}_l, \mathbf{c}_r \in \mathbb{R}^D$. These will be trained using gradients from an invertible transformation and resulting distribution defined in each Voronoi cell, which we discuss next.

**Invertible mapping onto subset**   Within the normalizing flow framework, defining a probability distribution on a bounded support is equivalent to defining an invertible transformation that maps from the unbounded support onto the appropriate support. Figure 2 illustrates the transformation of mapping onto a shaded region.

Given a cell $V_k$ for some $k \in \{1, \dots, K\}$, we construct an invertible mapping $f_k : \mathbb{R}^D \to V_k$ by following 2 steps. Let $\boldsymbol{x} \in \mathbb{R}^D$. First, if $\boldsymbol{x} = \boldsymbol{x_k}$, the anchor point of $V_k$, we simply set $f_k(\boldsymbol{x}) = \boldsymbol{x}$. Otherwise:

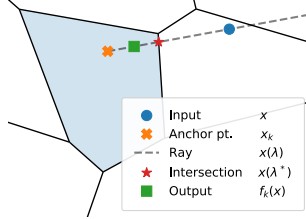

Figure 2: An illustration.

1. Determine where the ray starting from $\boldsymbol{x_k}$ in the direction of $\boldsymbol{x}$ intersects with the boundary of $V_k$.

First define the direction $\boldsymbol{\delta_k}(\boldsymbol{x}) \triangleq \frac{\boldsymbol{x} - \boldsymbol{x_k}}{\|\boldsymbol{x} - \boldsymbol{x_k}\|}$ and the ray $\boldsymbol{x}(\lambda) \triangleq \boldsymbol{x_k} + \lambda\boldsymbol{\delta_k}(\boldsymbol{x})$, with $\lambda > 0$. Since $V_k$ is convex, we can frame this as a linear programming problem.

$$\max \; \lambda \quad \text{s.t.} \quad \boldsymbol{x}(\lambda) \in \overline{V_k}, \lambda \geq 0 \quad (7)$$

where $\overline{V_k}$ is the closure of $V_k$. We discuss solving this efficiently in Section 3.1.

Let $\lambda^*$ be the solution. This solution exists if $V_k$ is bounded, which is always true due to the use of box constraints. Then $\boldsymbol{x}(\lambda^*)$ will be the point of intersection.

This first step solves for the farthest point in the Voronoi cell in the direction of $x$. Using this knowledge, we can now map all points that lie on this ray onto the Voronoi cell.

2. Apply an invertible transformation such that any point on the ray $\{x(\lambda) : \lambda > 0\}$ is mapped onto the line segment $\{x_k + \alpha(x(\lambda^*) - x_k) : \alpha \in (0, 1)\}$.

There are many possible choices for designing this transformation. An intuitive choice is to use a monotonic transformation of the relative distance from $x$ to the anchor point $x_k$.

$$f_k(x) \triangleq x_k + \alpha_k \left( \frac{\|x - x_k\|}{\|x(\lambda^*) - x_k\|} \right) (x(\lambda^*) - x_k) \tag{8}$$

where $\alpha_k$ is an appropriate *invertible* squashing function from $R^+$ to $(0, 1)$. In our experiments, we use $\alpha_k(h) = \text{softsign}(\gamma_k h)$ where $\gamma_k$ is a learned cell-dependent scale. Other choices should also work, such as a monotonic neural network; however, depending on the application, we may need to compute $\alpha_k^{-1}$ for computing the inverse mapping, so it's preferable to have an analytical inverse.

### 3.1  Remarks and Propositions

**Box constraints**   There can be continuity problems if a Voronoi cell is unbounded, as the solution to Equation (8) does not exist if $x(\lambda^*)$ diverges. Furthermore, when solving Equation (7), it can be difficult to numerically distinguish between an unbounded cell and one whose boundaries are very far away. It is for these reasons that we introduce box constraints (Equation 6) in the formulation of Voronoi cells which allows us to sidestep these issues for now.

**Solving for $\lambda^*$**   Equation (7) can be solved numerically, but this approach is prone to numerical errors and requires implicit differentiation through convex optimization solutions [1]. We instead note that the solution of Equation (7) can be expressed in closed form, since it is always going to be the intersection of the ray $x(\lambda)$ with the nearest linear constraint.

Let $a_i^\mathsf{T} x = b_i$ be the plane that represents one of the linear constraints in Equation (6), which are expressed in Equation (4). Let $\lambda_i$ be the intersection of this plane with the ray, *i.e.* it is the solution to $a_i^\mathsf{T} x(\lambda_i) = b_i$, then the solution is simply the smallest positive $\lambda_i$, which satisfies all the linear constraints:

$$\lambda^* = \min\{\lambda_i : \lambda_i > 0\}, \quad \text{where} \quad \lambda_i = \frac{b_i - a_i^\mathsf{T} x_k}{a_i^\mathsf{T} \delta_k(x)}. \tag{9}$$

There are a total of $K + 2D - 1$ linear constraints, including the Voronoi cell boundaries and box constraints, which can be computed fully in parallel. This also allows end-to-end differentiation of the mapping $f_k$ via automatic differentiation, providing the ability to learn the parameters of $f_k$, *i.e.* parameters of $\alpha_k$ and the Voronoi tessellation.

**Homeomorphism with continuous density**   We can show that $f_k$ is a bijection, and both $f_k$ and $f_k^{-1}$ are continuous. This allows us to use $f_k$ within the normalizing flows framework, as a mapping between a distribution $p_x$ defined on $R^D$ and the transformed distribution $p_z$ on $V_k$. Furthermore, the Jacobian is continuous almost everywhere. Proofs are in Appendix A.

*Proposition 1* The mapping $f_k : \mathbb{R}^D \to V_k$ as defined in the 2-step procedure is a homeomorphism.

*Proposition 2* If $p_x(x)$ is continuous, then the transformed density $p_z(f_k(x))$ is continuous a.e.

**Efficient computation of the log det Jacobian**   As $f_k$ is a mapping in $D$ dimensions, computing the log determinant Jacobian for likelihood computation can be costly and will scale poorly with $D$ if computed naïvely. Instead, we note that the Jacobian of $f_k$ is highly structured. Intuitively, because $f_k$ depends only on the direction $\delta_k(x)$ and the distance away from $x_k$, it only has two degrees of freedom regardless of $D$. In fact, the Jacobian of $f_k$ can be represented as a rank-2 update on a scaled identity matrix. This allows us to use the matrix determinant lemma to reformulate the log determinant Jacobian in a compute- and memory-efficient form. We summarize this in a proposition.

*Proposition 3* Let the transformation $f_k(x)$ and all intermediate quantities be as defined in Section 3 for some given input $x$. Then the Jacobian factorizes as

$$\frac{\partial f_k(x)}{\partial x} = cI + u_1 v_1^\mathsf{T} + u_2 v_2^\mathsf{T} \tag{10}$$

for some $c \in \mathbb{R}, u_i \in \mathbb{R}^D, v_i \in \mathbb{R}^D$, and its log determinant has the form

$$\log \left| \det \frac{\partial f_k(x)}{\partial x} \right| = \log |1 + w_{11}| + \log \left| 1 + w_{22} - \frac{w_{12} w_{21}}{1 + w_{11}} \right| + D \log c \tag{11}$$

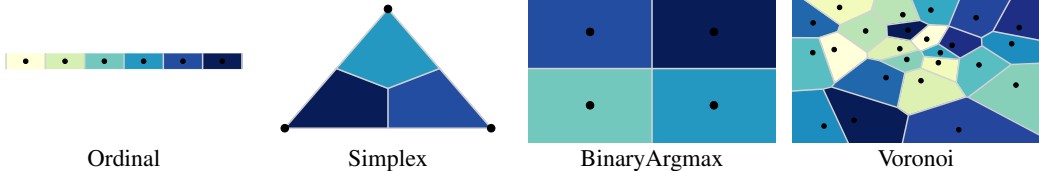

| Ordinal | Simplex | BinaryArgmax | Voronoi |

Figure 3: Existing dequantization methods can be seen as special cases of Voronoi dequantization with fixed anchor points. In additional to decoupling the dimension of the embedding space from the number of discrete values, the Voronoi dequantization can learn to model the similarities of discrete values through the positions and boundaries of their Voronoi cells.

where $w_{ij} = c^{-1} v_i^\mathsf{T} u_j$. This expression for the log determinant only requires inner products between vectors in $R^D$. To reduce notational clutter, exact formulas for $c, u_1, u_2, v_1, v_2$ are in Appendix A. All vectors used in computing Equation (11) are either readily available as intermediate quantities after computing $f_k(x)$ or are gradients of scalar functions and can be efficiently computed through reverse-mode automatic differentiation. The only operations on these vectors are dot products, and no large $D$-by-$D$ matrices are ever constructed. Compared to explicitly constructing the Jacobian matrix, this is more efficient in both compute and memory, and can readily scale to large values of $D$.

## 4    Application settings

We discuss two applications of our method to likelihood-based modeling of discrete and continuous data. The first is a new approach to dequantization which allows training a model of discrete data using density models that normally only act on continuous variables such as normalizing flows. Compared to existing dequantization methods [7, 18, 38], the Voronoi dequantization is not restricted to fixed dimensions and can benefit from learning similarities between discrete values. In fact, existing approaches can be seen as special cases of a Voronoi dequantization.

The second is a formulation of disjoint mixture models, where each component lies strictly within each subset of a Voronoi tessellation. To the best of our knowledge, disjoint mixture models have not been explored significantly in the literature and have not been successfully applied in more than a couple dimensions. Compared to an existing method [8], the Voronoi mixture model is not restricted to acting individually for each dimension.

**Voronoi dequantization**    Let $y$ be a discrete random variable with finite support $\mathcal{Y}$. Then we can use tessellation to define subsets for dequantizing $y$ as long as there are at least as many Voronoi cells—equivalently, anchors points—as the number of discrete values, *i.e.* $K \geq |\mathcal{Y}|$. By assigning each discrete value to a Voronoi cell, we can then define a dequantization model $q(\boldsymbol{x}|y)$ by first sampling from a base distribution $\boldsymbol{z} \sim q(\boldsymbol{z}|y)$ in $\mathbb{R}^D$ and then applying the mapping $\boldsymbol{x} = f_k(\boldsymbol{z})$ from Section 3 to construct a distribution over $V_k$. We can then obtain probabilities $q(\boldsymbol{x}|y)$ efficiently and train the dequantization alongside the density model $p(\boldsymbol{x})$ on $\mathbb{R}^D$.

Sampling from the model $p(y|\boldsymbol{x})$ is straightforward and deterministic after sampling $\boldsymbol{x}$. We can write $y = g(\boldsymbol{x})$ where $g$ is the set identification function satisfying $\boldsymbol{x} \in V_{g(\boldsymbol{x})}$. From Equation (4), it is easy to see that $g(\boldsymbol{x})$ is the nearest neighbor operation, $g(\boldsymbol{x}) = \arg\min_k \|\boldsymbol{x} - \boldsymbol{x_k}\|$.

We are free to choose the number of dimensions $D$, where a smaller $D$ assigns less space for each Voronoi cell induces dequantized distributions that are easier to fit, while a larger $D$ allows the anchor points and Voronoi cells more room to change over the course of training. When the anchor points are fixed to specific positions, we can recover the disjoint subsets used by prior methods as special cases. We illustrate this in Figure 3.

**Voronoi mixture models**    Let $\{V_k\}$ be a Voronoi tessellation of $\mathbb{R}^D$. Then a disjoint mixture model can be constructed by defining distributions on each Voronoi cell. Here we make use of the inverse mapping $f_k^{-1} : V_k \to \mathbb{R}^D$ (discussed in Appendix B) so that we only need to parameterize distributions over $\mathbb{R}^D$. Let $\boldsymbol{x}$ be a point in $\mathbb{R}^D$, our Voronoi mixture model assigns the density

$$\log p_{\mathrm{mix}}(\boldsymbol{x}) = \log p_{\mathrm{comp}}(f_{k=g(\boldsymbol{x})}^{-1}(\boldsymbol{x})|k = g(\boldsymbol{x})) + \log \left| \det \frac{\partial f_k^{-1}(\boldsymbol{x})}{\partial \boldsymbol{x}} \right| + \log p(k = g(\boldsymbol{x})) \quad (12)$$

where $p_{\text{comp}}$ can be any distribution over $\mathbb{R}^D$, including another disjoint mixture model. This can easily be composed with normalizing flow layers where, in addition to the change of variable due to $f_k^{-1}$, we also apply the change in density resulting from choosing one out of $K$ components.

# 5  Related Work

**Normalizing flows for discrete data**   Invertible mappings have been proposed for discrete data, where discrete values are effectively rearranged from a factorized distribution. In order to parameterize the transformation, Hoogeboom et al. [17], van den Berg et al. [47] use quantized ordinal transformations, while Tran et al. [45] takes a more general approach of using modulo addition on one-hot vectors. These approaches suffer from gradient bias due to the need to use discontinuous operations and do not have universality guarantees since it's unclear whether simple rearrangement is sufficient to transform any joint discrete distribution into a factorized distribution. In contrast, the dequantization approach provides a universality guarantee since the lower bound in Equation (2) is tight when $p(\boldsymbol{x})\mathbb{1}_{[\boldsymbol{x} \in A_y]} \propto q(\boldsymbol{x}|y)$, with a proportionality equal to $p(y)$.

**Dequantization methods**   Within the normalizing flows literature, the act of adding noise was originally used for ordinal data as a way to combat numerical issues [7, 46]. Later on, appropriate dequantization approaches have been shown to lower bound the log-likelihood of a discrete model [16, 44]. For non-ordinal data, many works have proposed simplex-based approaches. Early works on relaxations [20, 32] proposed continuous distribution on the simplex that mimic the behaviors of discrete random variables; however, these were only designed for the use with a Gumbel base distribution. Potapczynski et al. [38] extend this to a Gaussian distribution—although it is not hard to see this can work with any base distribution—by designing invertible transformations between $R^D$ and the probability simplex with $K$ vertices, with $D = K - 1$, where $K$ is the number of classes of a discrete random variable.

Intuitively, after fixing one of the logits, the softmax operation is an invertible transformation on the bounded domain $\Delta^{K-1} = \{\boldsymbol{x} \in \mathbb{R}^K : \sum_i^K \boldsymbol{x}_i = 1, \boldsymbol{x}_i > 0\}$. The $(K-1)$-simplex can then be broken into $K$ subsets, each corresponding to a particular discrete value.

$$A_k^{\text{simplex}} = \{\boldsymbol{x} \in \Delta^{K-1} : \boldsymbol{x}_k > \boldsymbol{x}_i \ \forall i \neq k\}. \tag{13}$$

More recently, Hoogeboom et al. [18] proposed ignoring the simplex constraint and simply use

$$A_k^{\text{argmax}} = \{\boldsymbol{x} \in \mathbb{R}^K : \boldsymbol{x}_k > \boldsymbol{x}_i \ \forall i \neq k\}, \tag{14}$$

which effectively increases the number of dimensions by one compared to the simplex approach. However, both approaches force the dimension of the continuous space $D$ to scale with $K$. In order to make their approach work when $K$ is large, [18] propose binarizing the data as a preprocessing step. In contrast, Voronoi dequantization has full flexibility in choosing $D$ regardless of $K$.

**Related models for discrete data**   Among other related methods, a recent work proposed normalizing flows on truncated supports [43] but had to resort to rejection sampling for training. Furthermore, their subsets are not disjoint by construction. Prior works [28, 31] also proposed removing the constraint that subsets are disjoint, and instead work with general mixture models with unbounded support, relying on the conditional model $p(y|\boldsymbol{x})$ being sufficiently weak so that the task of modeling is forced onto a flow-based prior. They have achieved good performance on a number of tasks, similar to general approaches that combine normalizing flows with variational inference [19, 33, 50]. However, they lose the computational savings and the deterministic decoder $p(y|\boldsymbol{x})$ gained from using disjoint subsets. On the other hand, quantization based on nearest neighbor have been used for learning discrete latent variables [34, 40], but no likelihoods are constructed, the boundaries are not explicitly differentiated through, and the model relies on training with heuristical updates.

**Disjoint mixture models**   The computational savings from using disjoint subsets was pointed out by Dinh et al. [8]. However, their method only works in each dimension individually. They transform samples using a linear spline, which is equivalent to creating subsets based on the knots of the spline and applying a linear transformation within each subset. Furthermore, certain parameterizations of the spline can lead to discontinuous density functions, whereas our disjoint mixture has a density function that is continuous almost everywhere (albeit exactly zero on the boundaries; see Section 7

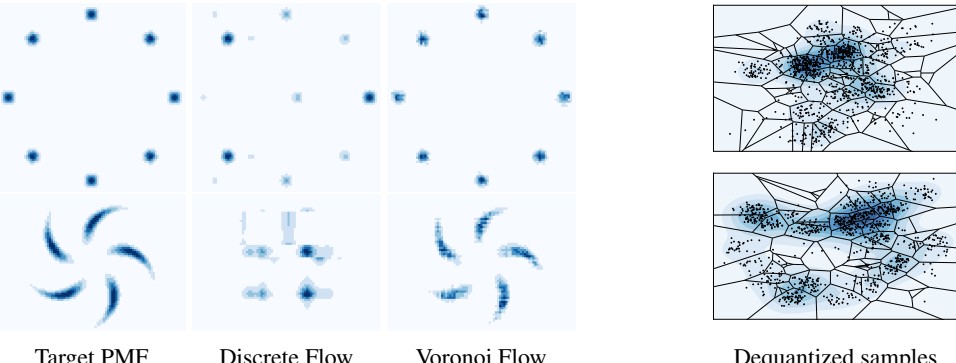

| Target PMF | Discrete Flow | Voronoi Flow | | Dequantized samples |

Figure 4: **Quantized 2D data.** Voronoi dequantization can model complex relations between discrete values. No knowledge of ordering is given to the models.

Figure 5: The model learns to cluster discrete values with similar probability values.

for more discussion). The use of monotonic splines have previously been combined with normalizing flows [10], but interestingly, since the splines in Dinh et al. [8] are not enforced to be monotonic, the full transformation is not bijective and acts like a disjoint mixture model. Ultimately, their experiments were restricted to two-dimensional syntheic data sets, and it remained an interesting research question whether disjoint mixture models can be successfully applied in high dimensions.

## 6 Experiments

We experimentally validate our semi-discrete approach of combining Voronoi tessellation with likelihood-based modeling on a variety of data domains: discrete-valued UCI data sets, itemset modeling, language modeling, and disjoint mixture modeling for continuous UCI data. The goal of these experiments is not to show state-of-the-art results in these domains but to showcase the relative merit of Voronoi tessellation compared to existing methods. For this reason, we just use simple coupling blocks with affine transformations [6, 7] as a base flow model in our experiments, unless stated otherwise. When comparing to Discrete Flows [45], we use their bipartite layers. Details regarding preprocessing for data sets can be found in Appendix C, and detailed experimental setup is in Appendix D. All tables, when shown, display standard deviations across three random seeds. Open source code is available at https://github.com/facebookresearch/semi-discrete-flow.

### 6.1 Quantized 2D data

We start by considering synthetic distributions over two discrete variables, to qualitatively compare against Discrete Flows [45]. Figure 4 shows the target probability mass functions, which are created by a 2D distribution where each dimension is quantized into 91 discrete values. Though the data is ordinal, no knowledge of ordering is given to the models. We see that Discrete Flows has trouble fitting these, because it is difficult to rearrange the target distribution into a factorized distribution. For our model, we dequantize each discrete variable with a Voronoi tessellation in $\mathbb{R}^2$. We then learn a flow model on the combined $\mathbb{R}^4$, parameterized by multilayer perceptrons (MLPs). In Figure 5 we visualize the learned Voronoi tessellation and samples from our model. The learned tessellation seems to group some of discrete values that occur frequently together, to reduce the number of modes.

### 6.2 Discrete-valued UCI data

We experiment with complex data sets where each discrete variable can have a varying number of classes. Furthermore, these discrete variables may have hidden semantics. To this end, we use unprocessed data sets from the UCI database [9]. The only pre-processing we perform is to remove variables that only have one discrete value. We then take 80% as train, 10% as validation, and 10% as the test set. Most of these data sets have a combination of both ordinal and non-ordinal variables, and we expect the non-ordinal variables to exhibit relations that are unknown (*e.g.* spatial correlations).

Table 1: **Discrete UCI data sets.** Negative log-likelihood results on the test sets in nats.

| Method | Connect4 | Forests | Mushroom | Nursery | PokerHands | USCensus90 |
|---|---|---|---|---|---|---|
| Voronoi Deq. | $12.92_{\pm0.07}$ | $14.20_{\pm0.05}$ | $9.06_{\pm0.05}$ | $9.27_{\pm0.04}$ | $19.86_{\pm0.04}$ | $24.19_{\pm0.12}$ |
| Simplex Deq. | $13.46_{\pm0.01}$ | $16.58_{\pm0.01}$ | $9.26_{\pm0.01}$ | $9.50_{\pm0.00}$ | $19.90_{\pm0.00}$ | $28.09_{\pm0.08}$ |
| BinaryArgmax Deq. | $13.71_{\pm0.04}$ | $16.73_{\pm0.17}$ | $9.53_{\pm0.01}$ | $9.49_{\pm0.00}$ | $19.90_{\pm0.01}$ | $27.23_{\pm0.02}$ |
| Discrete Flow | $19.80_{\pm0.01}$ | $21.91_{\pm0.01}$ | $22.06_{\pm0.01}$ | $9.53_{\pm0.01}$ | $19.82_{\pm0.03}$ | $55.62_{\pm0.35}$ |

Table 2: **Permutation-invariant discrete itemset modeling.**

| Model (Dequantization) | Retail (nats) | Accidents (nats) |
|---|---|---|
| CNF (Voronoi) | $9.44_{\pm2.34}$ | $7.81_{\pm2.84}$ |
| CNF (Simplex) | $24.16_{\pm0.21}$ | $19.19_{\pm0.01}$ |
| CNF (BinaryArgmax) | $10.47_{\pm0.42}$ | $6.72_{\pm0.23}$ |
| Determinantal Point Process | $20.35_{\pm0.05}$ | $15.78_{\pm0.04}$ |

Table 3: **Language modeling.**

| Dequantization | text8 (bpc) | enwik8 (bpc) |
|---|---|---|
| Voronoi ($D=2$) | $1.39_{\pm0.01}$ | $1.46_{\pm0.01}$ |
| Voronoi ($D=4$) | $1.37_{\pm0.00}$ | $1.41_{\pm0.00}$ |
| Voronoi ($D=6$) | $1.37_{\pm0.00}$ | $1.40_{\pm0.00}$ |
| Voronoi ($D=8$) | $1.36_{\pm0.00}$ | $1.39_{\pm0.01}$ |
| BinaryArgmax [18] | 1.38 | 1.42 |
| Ordinal [18] | 1.43 | 1.44 |

We see that Discrete Flows can be competitive with dequantization approaches, but can also fall short on more challenging data sets such as the USCensus90, the largest data set we consider with 2.7 million examples and 68 different discrete variables of varying types. For dequantization methods, simplex [38] and binary argmax [18] approaches are mostly on par. We do see a non-negligible gap in performance between these baselines and Voronoi dequantization for most of the data sets, likely due to the ability to learn semantically useful relations between the values of each discrete variable. For instance, the Connect4 data set contains positions of a two-player board game with the same name, which exhibit complex spatial dependencies between the board pieces, and the USCensus90 data set contains highly correlated variables and is often used to test clustering algorithms.

### 6.3  Permutation-invariant itemset modeling

An appeal of using normalizing flows in continuous space is the ability to incorporate invariance to specific group symmetries into the density model. For instance, this can be done by ensuring the ordinary differential equation is equivariant [2, 25, 42] in a continuous normalizing flow [4, 13] framework. Here we focus on invariance with respect to permutations, *i.e.* sets of discrete variables. This invariance cannot be explicitly modeled by Discrete Flows as they require an ordering or a bipartition of discrete variables. We preprocessed a data set of retail market basket data from an anonymous Belgian retail store [3] and a data set of anonymized traffic accident data [11], which contain sets of discrete variables each with 765 and 269 values, respectively. Without the binarization trick, simplex dequantization must use a large embedding dimensions and performs worse than a determinental point process baseline [26]. Meanwhile, our Voronoi dequantization has no problems staying competitive as its embedding space is freely decoupled from the number of discrete values.

### 6.4  Language modeling

Language modeling a widely used benchmark for discrete models [18, 28, 45]. Here we used the open source code provided by Hoogeboom et al. [18] with the exact same autoregressive flow model and optimizer setups. The only difference is replacing their binary argmax with our Voronoi dequantization. Results are shown in Table 3, where we tried out multiple embedding dimensions $D$. Generally, we find $D = 2$ to be too low and can stagnate training since the Voronoi cells are more constrained, while larger values of $D$ improve upon argmax dequantization.

### 6.5  Voronoi mixture models

We test disjoint mixture modeling with Voronoi tessellation on continuous data sets. Figure 6 shows results of training a simple normalizing flow on 2D data, which has trouble completely separating all the modes. Adding in a Voronoi mixture allows us to better fit these multimodal densities.

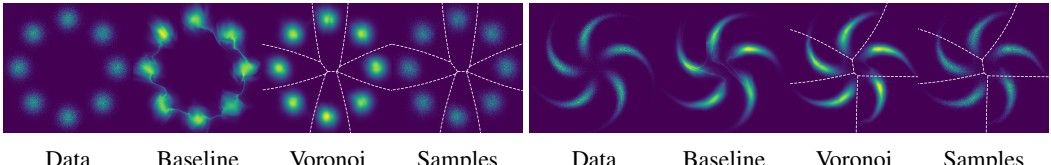

| Data | Baseline | Voronoi | Samples | Data | Baseline | Voronoi | Samples |

Figure 6: Tessellation is done in a transformed space; nonlinear boundaries are shown.

Table 4: **Disjoint mixture modeling.** NLL on the test sets in nats. *Baseline results from [13, 35].

| Method | POWER | GAS | HEPMASS | MINIBOONE | BSDS300 |
|---|---|---|---|---|---|
| Real NVP* | $-0.17_{\pm 0.01}$ | $-8.33_{\pm 0.07}$ | $18.71_{\pm 0.01}$ | $13.55_{\pm 0.26}$ | $-153.28_{\pm 0.89}$ |
| MAF* | $-0.24_{\pm 0.01}$ | $-10.08_{\pm 0.01}$ | $17.73_{\pm 0.01}$ | $12.24_{\pm 0.22}$ | $-154.93_{\pm 0.14}$ |
| FFJORD* | $-0.46_{\pm 0.01}$ | $-8.59_{\pm 0.12}$ | $14.92_{\pm 0.08}$ | $10.43_{\pm 0.22}$ | $-157.40_{\pm 0.19}$ |
| Base Coupling Flow | $-0.44_{\pm 0.01}$ | $-11.75_{\pm 0.02}$ | $16.78_{\pm 0.08}$ | $10.87_{\pm 0.06}$ | $-155.14_{\pm 0.04}$ |
| Voronoi Disjoint Mixture | $-0.52_{\pm 0.01}$ | $-12.63_{\pm 0.05}$ | $16.16_{\pm 0.01}$ | $10.24_{\pm 0.14}$ | $-156.59_{\pm 0.14}$ |

To test Voronoi mixture models on higher dimensions, we apply our approach to data sets preprocessed by Papamakarios et al. [35], which range from 6 to 63 dimensions. We add a disjoint mixture component after a few layers of normalizing flows, with each component modeled by a normalizing flow conditioned on a embedding of the component index. The number of mixture components ranges from 16 to 64, with complete sweeps and hyperparameters detailed in Appendix D. For comparison, we also show results from the baseline coupling flow that uses the same number of total layers as the disjoint mixture, as well as a strong but computationally costly density model FFJORD [13]. From Table 4, we see that the disjoint mixture model approach allows us to increase complexity quite a bit, with almost no additional cost compared to the baseline flow model.

## 7    Conclusion and Discussion

We combine Voronoi tessellation with normalizing flows to construct a new invertible transformation that has learnable discrete structure. This acts as a learnable mapping between discrete and continuous distributions. We propose two applications of this method: a Voronoi dequantization that maps discrete values into a learnable convex polytope, and a Voronoi mixture modeling approach that has around the same compute cost as a single component. We showcased the relative merit of our approach across a range of data modalities with varying hidden structure. Below, we discuss some limitations and possible future directions.

**Diminishing density on boundaries.** The distributions within each Voronoi cell necessarily go to zero on the boundary between cells due to the use of a homeomorphism from an unbounded domain. This is a different and undesirable property as opposed to the partitioning method used by Dinh et al. [8]. However, alternatives could require more compute cost in the form of solving normalization constants. Balancing expressiveness and compute cost is a delicate problem that sits at the core of probabilistic machine learning.

**Design of homeomorphisms and tessellations.** Our proposed homeomorphism is simple and computationally scalable, but this comes at the cost of smoothness and expressiveness. As depicted in Figure 1, the transformed density has a non-smooth probability density function. This non-smoothness exists when the ray intersects with multiple boundaries. This may make optimization more difficult as the gradients of the log-likelihood objective can be discontinuous. Additionally, our use of Euclidean distance can become problematic in high dimensions, as this metric can cause almost all points to be nearly equidistant, resulting in a behavior where all points lie seemingly very close to the boundary. Improvements on the design of homeomorphisms to bounded domains could help alleviate these problems. Additionally, more flexible tessellations—such as the Laguerre tessellation—and additional concepts from semi-discrete optimal transport [14, 27, 37] may be adapted to improve semi-discrete normalizing flows.

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
