# A  Proofs

*Proposition 1* The mapping $f_k : \mathbb{R}^D \to V_k$ as defined in the 2-step procedure is a homeomorphism.

*Proof.* Let $x \in \mathbb{R}^D$ and $x_k$ be a given anchor point corresponding to a Voronoi cell $V_k$. If $x \neq x_k$, then $x$ is uniquely represented by the tuple $(\Delta, \delta)$ where $\Delta = \|x - x_k\|$ and $\delta = \frac{x - x_k}{\|x - x_k\|}$, since $x = x_k + \Delta \delta$. Since $\delta$ uniques defines the ray $\{x_k + \lambda \delta; \lambda > 0\}$ and $x(\lambda^*)$, then because $\alpha_k$ in Equation (8) is a bijection in $\Delta$, $f_k$ is a bijection. For $x \neq x_k$, the continuity of $f_k$ follows from Rudin et al. [41, Theorem 4.7] since $\Delta$ and $x(\lambda^*)$ are continuous in $x$ and $\alpha_k$ is continuous in $\Delta$. Then since $f_k(x) \to x_k$ as $x \to x_k$ from all directions, this justifies the choice of setting $f_k(x) = x_k$ when $x = x_k$. Finally, by the invariance of domain theorem, since $V_k$ is an open set in $\mathbb{R}^D$, $f_k$ is an open map and the inverse $f_k^{-1}$ is continuous, and we can conclude $f$ is a homeomorphism between $\mathbb{R}^D$ and $V_k$. $\qquad\square$

*Proposition 2* If $p_x(x)$ is continuous, then the transformed density $p_z(f_k(x))$ is continuous a.e.

*Proof.* See proof of Proposition 3 below for the form of the Jacobian. For the case where $x \neq x_k$, all quantities in Equations (15) to (22) are continuous with respect to $x$. Hence the Jacobian $\frac{\partial f_k(x)}{\partial x}$ is continuous, and since it is always full-rank, then the composition $\left| \det \frac{\partial f_k(x)}{\partial x} \right|$ is continuous [41, Theorem 4.7] and so is the product $p_x(x) \left| \det \frac{\partial f_k(x)}{\partial x} \right|$ [41, Theorem 4.9]. $\qquad\square$

*Proposition 3* Let the transformation $f_k(x)$ and all intermediate quantities be as defined in Section 3 for some given input $x$. Then the Jacobian factorizes as

$$\frac{\partial f_k(x)}{\partial x} = cI + u_1 v_1^\mathsf{T} + u_2 v_2^\mathsf{T} \tag{10}$$

for some $c \in \mathbb{R}, u_i \in \mathbb{R}^D, v_i \in \mathbb{R}^D$, and its log determinant has the form

$$\log \left| \det \frac{\partial f_k(x)}{\partial x} \right| = \log |1 + w_{11}| + \log \left| 1 + w_{22} - \frac{w_{12} w_{21}}{1 + w_{11}} \right| + D \log c \tag{11}$$

*where*

$$\Delta \triangleq \|x - x_k\| \tag{15}$$

$$c \triangleq \alpha_k(\tilde{\Delta}) \lambda^* \Delta^{-1} \tag{16}$$

$$u_1 \triangleq \left[ \alpha_k(\tilde{\Delta}) \Delta^{-1} - \frac{\partial \alpha_k(\tilde{\Delta})}{\partial \tilde{\Delta}} \right] \delta_k(x) \tag{17}$$

$$v_1 \triangleq \frac{\partial \lambda^*}{\partial \delta_k(x)} \tag{18}$$

$$u_2 \triangleq \left[ \left( \frac{\partial \alpha_k(\tilde{\Delta})}{\partial \tilde{\Delta}} \right) - \alpha_k(\tilde{\Delta}) \lambda^* \Delta^{-1} - \alpha_k(\tilde{\Delta}) \Delta^{-1} \left( \left( \frac{\partial \lambda^*}{\partial \delta_k(x)} \right)^\mathsf{T} \delta_k(x) \right) \right. \tag{19}$$

$$\left. + \left( \frac{\partial \alpha_k(\tilde{\Delta})}{\partial \tilde{\Delta}} \right) \left( \left( \frac{\partial \lambda^*}{\partial \delta_k(x)} \right)^\mathsf{T} \delta_k(x) \right) \right] \delta_k(x) \tag{20}$$

$$v_2 \triangleq \delta_k(x) \tag{21}$$

$$w_{ij} \triangleq c^{-1} v_i^\mathsf{T} u_j, \text{ for } i, j \in \{1, 2\} \tag{22}$$

*Proof.* We first write the Jacobian of $f_k$ in the form of $cI + u_1 v_1^\mathsf{T} + u_2 v_2^\mathsf{T}$ where $c$ is a scalar, and $u_1, u_2, v_1, v_2$ are vectors of size $D$. Define the shorthand $\Delta \triangleq \|x - x_k\|$, $\Delta^* \triangleq \|x(\lambda^*) - x_k\|$, and $\tilde{\Delta} \triangleq \Delta / \Delta^*$. To simplify notation, we use the short-hands $\alpha_k = \alpha_k(\tilde{\Delta})$, $\delta_k = \delta_k(x)$. Then the

Jacobian follows

$$\frac{\partial f_k(x)}{\partial x} \tag{23}$$

$$= \frac{\partial}{\partial x}\left(x_k + \alpha_k(x(\lambda^*) - x_k)\right) \tag{24}$$

$$= (x(\lambda^*) - x_k)\left(\frac{\partial \alpha_k}{\partial \tilde{\Delta}}\right)\left(\frac{\partial \tilde{\Delta}}{\partial x}\right)^{\mathsf{T}} + \alpha_k\left(\frac{\partial x(\lambda^*)}{\partial x}\right) \tag{25}$$

$$= (x(\lambda^*) - x_k)\left(\frac{\partial \alpha_k}{\partial \tilde{\Delta}}\right)\left(\frac{1}{\Delta^*}\delta_k^{\mathsf{T}} - \frac{\Delta}{(\Delta^*)^2}(x(\lambda^*) - x_k)^{\mathsf{T}}\frac{\partial x(\lambda^*)}{\partial x}\right) + \alpha_k\left(\frac{\partial x(\lambda^*)}{\partial x}\right) \tag{26}$$

$$= \left(\frac{\partial \alpha_k}{\partial \tilde{\Delta}}\right)\delta_k\delta_k^{\mathsf{T}} - \left(\frac{\partial \alpha_k}{\partial \tilde{\Delta}}\right)\Delta\delta_k\delta_k^{\mathsf{T}}\frac{\partial x(\lambda^*)}{\partial x} + \alpha_k\left(\frac{\partial x(\lambda^*)}{\partial x}\right) \tag{27}$$

$$= \left(\frac{\partial \alpha_k}{\partial \tilde{\Delta}}\right)\delta_k\delta_k^{\mathsf{T}} + \left(\alpha_k - \left(\frac{\partial \alpha_k}{\partial \tilde{\Delta}}\right)\Delta\delta_k\delta_k^{\mathsf{T}}\right)\left(\frac{\partial x(\lambda^*)}{\partial x}\right) \tag{28}$$

$$= \left(\frac{\partial \alpha_k}{\partial \tilde{\Delta}}\right)\delta_k\delta_k^{\mathsf{T}} + \left(\alpha_k - \left(\frac{\partial \alpha_k}{\partial \tilde{\Delta}}\right)\Delta\delta_k\delta_k^{\mathsf{T}}\right)\left(\lambda^*\Delta^{-1}I - \lambda^*\Delta^{-1}\delta_k\delta_k^{\mathsf{T}} + \delta_k\left(\frac{\partial \lambda^*}{\partial \delta_k}\right)^{\mathsf{T}}\left(\frac{\partial \delta_k}{\partial x}\right)\right) \tag{29}$$

$$= \left(\frac{\partial \alpha_k}{\partial \tilde{\Delta}}\right)\delta_k\delta_k^{\mathsf{T}} + \left(\alpha_k - \left(\frac{\partial \alpha_k}{\partial \tilde{\Delta}}\right)\Delta\delta_k\delta_k^{\mathsf{T}}\right)\left(\lambda^*\Delta^{-1}I - \lambda^*\Delta^{-1}\delta_k\delta_k^{\mathsf{T}} + \Delta^{-1}\delta_k\left(\frac{\partial \lambda^*}{\partial \delta_k}\right)^{\mathsf{T}}\left(I - \delta_k\delta_k^{\mathsf{T}}\right)\right) \tag{30}$$

$$= \left(\frac{\partial \alpha_k}{\partial \tilde{\Delta}}\right)\delta_k\delta_k^{\mathsf{T}} + \left(\alpha_k - \left(\frac{\partial \alpha_k}{\partial \tilde{\Delta}}\right)\Delta\delta_k\delta_k^{\mathsf{T}}\right)\left(\lambda^*\Delta^{-1}I - \lambda^*\Delta^{-1}\delta_k\delta_k^{\mathsf{T}} + \Delta^{-1}\delta_k\left(\frac{\partial \lambda^*}{\partial \delta_k}\right)^{\mathsf{T}} - \Delta^{-1}\left(\left(\frac{\partial \lambda^*}{\partial \delta_k}\right)^{\mathsf{T}}\delta_k\right)\delta_k\delta_k^{\mathsf{T}}\right) \tag{31}$$

$$= \left(\frac{\partial \alpha_k}{\partial \tilde{\Delta}}\right)\delta_k\delta_k^{\mathsf{T}} + \alpha_k\lambda^*\Delta^{-1}I - \alpha_k\lambda^*\Delta^{-1}\delta_k\delta_k^{\mathsf{T}} + \alpha_k\Delta^{-1}\delta_k\left(\frac{\partial \lambda^*}{\partial \delta_k}\right)^{\mathsf{T}} - \alpha_k\Delta^{-1}\left(\left(\frac{\partial \lambda^*}{\partial \delta_k}\right)^{\mathsf{T}}\delta_k\right)\delta_k\delta_k^{\mathsf{T}} \tag{32}$$

$$\quad - \left(\frac{\partial \alpha_k}{\partial \tilde{\Delta}}\right)\lambda^*\delta_k\delta_k^{\mathsf{T}} + \left(\frac{\partial \alpha_k}{\partial \tilde{\Delta}}\right)\lambda^*\left(\delta_k^{\mathsf{T}}\delta_k\right)\delta_k\delta_k^{\mathsf{T}} - \left(\frac{\partial \alpha_k}{\partial \tilde{\Delta}}\right)\left(\delta_k^{\mathsf{T}}\delta_k\right)\delta_k\left(\frac{\partial \lambda^*}{\partial \delta_k}\right)^{\mathsf{T}} \tag{33}$$

$$\quad + \left(\frac{\partial \alpha_k}{\partial \tilde{\Delta}}\right)\left(\delta_k^{\mathsf{T}}\delta_k\right)\left(\left(\frac{\partial \lambda^*}{\partial \delta_k}\right)^{\mathsf{T}}\delta_k\right)\delta_k\delta_k^{\mathsf{T}} \tag{34}$$

$$= \alpha_k\lambda^*\Delta^{-1}I + \left[\alpha_k\Delta^{-1} - \left(\frac{\partial \alpha_k}{\partial \tilde{\Delta}}\right)\left(\delta_k^{\mathsf{T}}\delta_k\right)\right]\delta_k\left(\frac{\partial \lambda^*}{\partial \delta_k}\right)^{\mathsf{T}} \tag{35}$$

$$\quad + \left[\left(\frac{\partial \alpha_k}{\partial \tilde{\Delta}}\right) - \alpha_k\lambda^*\Delta^{-1} - \alpha_k\Delta^{-1}\left(\left(\frac{\partial \lambda^*}{\partial \delta_k}\right)^{\mathsf{T}}\delta_k\right) - \lambda^*\left(\frac{\partial \alpha_k}{\partial \tilde{\Delta}}\right)\left(\delta_k^{\mathsf{T}}\delta_k - 1\right) + \left(\frac{\partial \alpha_k}{\partial \tilde{\Delta}}\right)\left(\delta_k^{\mathsf{T}}\delta_k\right)\left(\left(\frac{\partial \lambda^*}{\partial \delta_k}\right)^{\mathsf{T}}\delta_k\right)\right]\delta_k\delta_k^{\mathsf{T}} \tag{36}$$

$$= \alpha_k\lambda^*\Delta^{-1}I + \left[\alpha_k\Delta^{-1} - \left(\frac{\partial \alpha_k}{\partial \tilde{\Delta}}\right)\right]\delta_k\left(\frac{\partial \lambda^*}{\partial \delta_k}\right)^{\mathsf{T}} \tag{37}$$

$$\quad + \left[\left(\frac{\partial \alpha_k}{\partial \tilde{\Delta}}\right) - \alpha_k\lambda^*\Delta^{-1} - \alpha_k\Delta^{-1}\left(\left(\frac{\partial \lambda^*}{\partial \delta_k}\right)^{\mathsf{T}}\delta_k\right) + \left(\frac{\partial \alpha_k}{\partial \tilde{\Delta}}\right)\left(\left(\frac{\partial \lambda^*}{\partial \delta_k}\right)^{\mathsf{T}}\delta_k\right)\right]\delta_k\delta_k^{\mathsf{T}} \tag{38}$$

Now that $\frac{\partial f_k(x)}{\partial x}$ is in the form of $cI + u_1 v_1^{\mathsf{T}} + u_2 v_2^{\mathsf{T}}$, where

$$c \triangleq \alpha_k(\tilde{\Delta})\lambda^*\Delta^{-1} \tag{39}$$

$$u_1 \triangleq \left[\alpha_k(\tilde{\Delta})\Delta^{-1} - \frac{\partial \alpha_k(\tilde{\Delta})}{\partial \tilde{\Delta}}\right]\boldsymbol{\delta_k(x)} \tag{40}$$

$$v_1 \triangleq \frac{\partial \lambda^*}{\partial \boldsymbol{\delta_k(x)}} \tag{41}$$

$$u_2 \triangleq \left[\left(\frac{\partial \alpha_k(\tilde{\Delta})}{\partial \tilde{\Delta}}\right) - \alpha_k(\tilde{\Delta})\lambda^*\Delta^{-1} - \alpha_k(\tilde{\Delta})\Delta^{-1}\left(\left(\frac{\partial \lambda^*}{\partial \boldsymbol{\delta_k(x)}}\right)^{\mathsf{T}}\boldsymbol{\delta_k(x)}\right) + \left(\frac{\partial \alpha_k(\tilde{\Delta})}{\partial \tilde{\Delta}}\right)\left(\left(\frac{\partial \lambda^*}{\partial \boldsymbol{\delta_k(x)}}\right)^{\mathsf{T}}\boldsymbol{\delta_k(x)}\right)\right]\boldsymbol{\delta_k(x)} \tag{42}$$

$$v_2 \triangleq \boldsymbol{\delta_k(x)} \tag{43}$$

Applying the matrix determinant lemma twice, we can show that

$$\det(cI + u_1 v_1^{\mathsf{T}} + u_2 v_2^{\mathsf{T}}) = (1 + v_2^{\mathsf{T}}(cI + u_1 v_1^{\mathsf{T}})^{-1}u_2)\det(cI + u_1 v_1^{\mathsf{T}}) \tag{44}$$

$$= \left[\left(1 + c^{-1}v_2^{\mathsf{T}}u_2 - \frac{c^{-2}}{1 + c^{-1}v_1^{\mathsf{T}}u_1}\right)v_2^{\mathsf{T}}u_1 v_1^{\mathsf{T}}u_2\right](1 + c^{-1}v_1^{\mathsf{T}}u_1)c^D \tag{45}$$

We simplify this by defining the scaled dot products,

$$w_{ij} \triangleq c^{-1}v_i^{\mathsf{T}}u_j, \text{ for } i, j \in \{1, 2\}. \tag{46}$$

Then
$$\log \left| \det \frac{\partial f_k(x)}{\partial x} \right| = \log |1 + w_{11}| + \log \left| 1 + w_{22} - \frac{w_{12} w_{21}}{1 + w_{11}} \right| + D \log c \tag{47}$$

$\square$

Intermediate steps above used the following gradient identities.

$$
\begin{aligned}
\frac{\partial \boldsymbol{\delta_k}(\boldsymbol{x})}{\partial x} &= \frac{\partial}{\partial x}(x - x_k)\Delta^{-1} \\
&= \Delta^{-1} I + (x - x_k)\left( \frac{\partial}{\partial x}\left(\Delta^2\right)^{-\frac{1}{2}} \right)^{\mathsf{T}} \\
&= \Delta^{-1} I + (x - x_k)\left( \frac{-1}{2\Delta^3} \right)\left( \frac{\partial \Delta^2}{\partial x} \right)^{\mathsf{T}} \\
&= \Delta^{-1} I + (x - x_k)\left( \frac{-1}{2\Delta^3} \right) 2 \left( x - x_k \right)^{\mathsf{T}} \\
&= \Delta^{-1} \left( I - \boldsymbol{\delta_k}(\boldsymbol{x})\boldsymbol{\delta_k}(\boldsymbol{x})^{\mathsf{T}} \right)
\end{aligned}
\tag{48}
$$

$$\frac{\partial \Delta}{\partial x} = \frac{\partial \Delta}{\partial x} = \frac{\partial}{\partial x}\left(\Delta^2\right)^{\frac{1}{2}} = \frac{1}{\Delta}(x - x_k) = \boldsymbol{\delta_k}(\boldsymbol{x}) \tag{49}$$

## B   The inverse mapping

We use the inverse mapping $f_k^{-1} : V_k \to \mathbb{R}^D$ for training disjoint mixture models, so we next describe how to compute this. Let $\boldsymbol{z} = f_k(\boldsymbol{x})$.

Conveniently, since both $\boldsymbol{x}$ and $\boldsymbol{z}$ lie on the ray $\{\boldsymbol{x}(\lambda) : \lambda > 0\}$, we know $\boldsymbol{\delta_k}(\boldsymbol{x}) = \boldsymbol{\delta_k}(\boldsymbol{z})$. So the first step is the same as the forward procedure: we solve for $\lambda^*$ and $\boldsymbol{x}(\lambda^*)$. Following this, we then recover $\boldsymbol{x}$ by inverting Step 2 of the forward procedure.

This inverse transformation is given by

$$\tilde{\alpha} = \frac{\boldsymbol{z} - \boldsymbol{x_k}}{\boldsymbol{x}(\lambda^*) - \boldsymbol{x_k}} \tag{50}$$

$$\tilde{\Delta} = \alpha_k^{-1}(\tilde{\alpha}_1) \tag{51}$$

$$\Delta = \tilde{\Delta} \, \|\boldsymbol{x}(\lambda^*) - \boldsymbol{x_k}\| \tag{52}$$

$$x = \Delta \boldsymbol{\delta_k}(\boldsymbol{z}) + \boldsymbol{x_k} \tag{53}$$

Equation (50) is an element-wise division. Since $\tilde{\alpha}$ will the same in all dimensions, we can simply pick a dimension in Equation (51). In our experiments, the inverse $\alpha_k^{-1}$ can be computed analytically, though since it is just a scalar function, simple methods like bisection can also work when the inverse is not known analytically. Lastly, Equation (53) follows from the observation that $\boldsymbol{\delta_k}(\boldsymbol{x}) = \boldsymbol{\delta_k}(\boldsymbol{z})$.

**Log determinant of the inverse.**   We can also use Proposition 3 to compute the log determinant of the inverse transform without needing to recompute $f_k(x)$. The only difference is a sign: $\log \left| \det \frac{\partial f_k^{-1}(z)}{\partial z} \right| = -\log \left| \det \frac{\partial f_k(x)}{\partial x} \right|$. The required quantities, $\Delta$, $\boldsymbol{x}(\lambda^*)$, and $\boldsymbol{\delta_k}(\boldsymbol{x})$, are readily available after computing $x = f_k^{-1}(z)$. The gradients with respect to quantities of $x$ can be expressed using gradients with respect to quantities of $z$,

$$\frac{\partial \alpha_k(\Delta)}{\partial \Delta} = \left( \frac{\partial \alpha_k^{-1}(\tilde{\alpha})}{\partial \tilde{\alpha}} \right)^{-1} \qquad \text{and} \qquad \frac{\partial \lambda^*}{\partial \boldsymbol{\delta_k}(\boldsymbol{x})} = \frac{\partial \lambda^*}{\partial \boldsymbol{\delta_k}(\boldsymbol{z})}, \tag{54}$$

which are accessible through automatic differentiation.

## C   Data sets

### C.1   UCI Data sets

The main preprocessing we did was to (i) remove the "label" attribute from each data set, and (ii) remove attributes that only ever take on one value. Apart from this, the `USCensus90` data set contains a unique identifier for each row, which was removed. Descriptions for all data set are below.

**Connect4** [webpage]   This data set contains all legal 8-ply positions in the game of Connect Four in which neither player has won yet, and in which the next move is not forced. The original task was to predict which player would win, which has been removed during preprocessing. There are a total **42** discrete variables (one for each location on the board), each with **3** possible discrete values (taken by player 1, taken by player 2, blank). This data set was randomly split into 54045 training examples, 6755 validation examples, and 6757 test examples.

**Forests** [webpage]   This data set contains cartographic variables regarding forests including four wilderness areas located in the Roosevelt National Forest of northern Colorado. These areas represent forests with minimal human-caused disturbances. The original task was to predict the forest cover type, which has been removed during preprocessing. There are a total of **54** discrete variables, with **10** being the highest number of discrete values. This data set was randomly split into 464809 training examples, 58101 validation examples, and 58102 test examples.

**Mushroom** [webpage]   This data set includes descriptions of hypothetical samples corresponding to 23 species of gilled mushrooms in the Agaricus and Lepiota Family. The original task was to predict whether each species is edible, which has been removed during preprocessing. There are a total of **21** discrete variables, with **12** being the highest number of discrete values. This data set was randomly split into 6499 training examples, 812 validation examples, and 813 test examples.

**Nursery** [webpage]   This data set contains attributes of applicants to nursery schools, during a period when there was excessive enrollment to these schools in Ljubljana, Slovenia, and the rejected applications frequently needed an objective explanation. All data have been completely anonymized. The original task was to predict whether an applicant would be recommended for acceptance by hierarchical decision model, which has been removed during preprocessing. There are a total of **8** discrete variables, with **5** being the highest number of discrete values. This data set was randomly split into 10367 training examples, 1296 validation examples, and 1297 test examples.

**PokerHands** [webpage]   This data set contains poker hands consisting of five playing cards drawn from a standard deck of 52. Each card is described using two attributes (suit and rank), for a total of 10 predictive attributes. There is one Class attribute that describes the "poker hand". The original task was to predict the poker hand class (pairs, full house, royal flush, etc.), which has been removed during preprocessing. There are a total of **10** discrete variables, with **13** being the highest number of discrete values. This data set was randomly split into 820008 training examples, 102501 validation examples, and 102501 test examples.

**USCensus90** [webpage]   This data set contains a portion of the data collected as part of the 1990 census in the United States, with the data completely anonymized. There are a total of **68** discrete variables, with **18** being the highest number of discrete values. This data set was randomly split into 2212456 training examples, 122914 validation examples, and 122915 test examples.

### C.2   Itemset Data sets

These data sets were taken from the Frequent Itemset Mining Data set Repository [webpage]. Each row is interpreted as a set of items with no emphasis on the ordering of items.

**Retail** [3]   This data set contains anonymized retail market basket data from an anonymous Belgian retail store. We first removed rows with less than 4 items, then randomly sampled a subset of 4 items for every row. Items that appear in less 300 rows were dropped from the data set. The final preprocessed data set contains **765** distinct items. This data set was randomly split into 24280 training examples, 3035 validation examples, and 3036 test examples.

**Accidents** [11]  This data set contains contains anonymized traffic accident data. Data on traffic accidents are obtained from the National Institute of Statistics (NIS) for the region of Flanders (Belgium) for the period 1991-2000. We first removed rows with less than 4 items, then randomly sampled a subset of 4 items for every row. This subsampling occurred 10 times if a row has 16 or more items, 5 times if the row has 8 to 15 items, and once if the row has 4-7 items. Items that appear in less 300 rows were dropped from the data set. The final preprocessed data set contains **213** distinct items. This data set was randomly split into 270129 training examples, 33766 validation examples, and 33767 test examples.

## D  Experimental Details

All experiments were run on a single NVIDIA V100 GPU. Detailed hyperparameter sweeps are below. We used the validation set to choose hyperparameters as well as to perform early stopping.

**2D synthetic data sets**  Continuous data sets were quantized into 91 bins for each coordinate. For Voronoi dequantization, we dequantized each coordinate into an embedding space of 2 dimensions, with 91 Voronoi cells. The dequantization model is parameterized by 4 layers of coupling blocks, each with a 2 hidden layer MLP with 256 hidden units each, where the Swish activation function was used [39]. The flow model is similarly parameterized but with 16 layers of coupling blocks. Each block alternated between 4 different partitioning schemes: maksing out the first half, masking out the second half, masking out the odd indices, and masking out the even indices. We trained with the `Adam` optimizer [22] with a learning rate of `1e-3`.

**Discrete-valued UCI data sets**  For Voronoi dequantization, we dequantized each coordinate into an embedding space of 4 or 6 dimensions, with the number of Voronoi cells set to the highest number of discrete values over all discrete variables. The dequantization model is parameterized by 4 layers of coupling blocks, each with a 2 hidden layer MLP with 256, 512, or 1024 hidden units each, where the Swish activation function was used [39]. The flow model is similarly parameterized but with 16 or 32 layers of coupling blocks. Each block alternated between 4 different partitioning schemes: maksing out the first half, masking out the second half, masking out the odd indices, and masking out the even indices. We trained with the `Adam` optimizer [22] with a learning rate sweep over {`1e-3`, `5e-4`, `1e-4`}.

**Itemset data sets**  For Voronoi dequantization, we dequantized each coordinate into an embedding space of 6 dimensions, with the number of Voronoi cells set to the number of items in the data set. We used a continuous normalizing flow (CNF) with the ordinary differential equation (ODE) defined using a Transformer archiecture and a $L_2$-distance based multihead attention layer [21] and the GeLU activation function [15]. No positional embeddings were provided to the model to ensure the model is equivariant to permutations. We composed 12 CNF layers, each defined using a Transformer model that has 2 or 3 layers of alternating multihead attention and fully connected residual connections. To solve the ODE and train our model, we used the `dopri5` solver from the torchdiffeq library [5] with `atol=rtol=1e-5`. We trained with the `AdamW` optimizer [30, 49] with a learning rate of `1e-3` and weight decay of `1e-6`. For the Voronoi dequantization, we set $D$=6 for both data sets, though it may be possible to improve performance by tuning $D$.

**Character-level language modeling**  We used the provided hyperparameters from the open source repository [URL]. Some parts of code had to be adapted for our usage, but model architecture and optimizer remained largely the same.

**Continuous-valued UCI data sets**  Disjoint mixture modeling is the inverse of a dequantization method. We use a flow model which maps the data nonlinearly, $z = f(x)$. Then a Voronoi mixture model partitions the space and assigns each $z$ an index through the set identification function $k = g(z)$. The probabilities $p(k)$ are parameterized through a softmax and learned. We then apply the inverse transformation from Section 3 to map from $V_k$ to $\mathbb{R}^D$. A conditional normalizing flow is then used to model each mixture model $p(z|k)$. We set the base distribution to be a Gaussian with standard deviation 0.2, as this helps concentrate the density around the anchor points at initialization. We tuned the number of flow layers before and after the mixture layer, with the total number of layers in {16, 32, 48, 64}. We also tuned the number of mixtures in {8, 16, 32, 64, 128}. Each coupling block

uses a neural network with 3 hidden layers of dimension $64$ with either the GeLU or Swish [39] activation function. We trained with a batch size of $1048$ and the `Adam` optimizer [22] with a learning rate sweep over $\{$`1e-3, 5e-3`$\}$.