# OpenReview forum: "Semi-Discrete Normalizing Flows through Differentiable Tessellation"
_NeurIPS.cc/2022/Conference — NeurIPS 2022 Accept_

### Official Review · Reviewer_xCLG · 2022-07-09

**Rating:** 8
**Confidence:** 3
**Soundness:** 4 excellent
**Presentation:** 4 excellent
**Contribution:** 4 excellent

**Summary:**

The paper applies Voronoi tessellation to allow the application of normalizing flows on discrete data. The main contributions are Voronoi dequantization, which is a new mapping method between discrete values and continuous space, and a disjoint mixture model, which uses an invertible mapping between unbounded support and convex polytopes. Experimental result on a variety of datasets shows that the proposed method outperforms existing baselines.

**Questions:**

- Since the experiments mainly compared with other flow models, it would be interesting to see how the proposed methods compare with state-of-the-art methods. Is normalizing flow outperformed by other methods even with the Voronoi dequantization and mixture modeling, or can it actually have better performance?

**Limitations:**

The authors have discussed limitations, along with possible ways to address them, in the conclusion.


**Strengths And Weaknesses:**

Strengths:
- The Voronoi tessellation idea is simple, novel, and intuitive. Additionally, it addresses an important problem of modeling discrete data with normalizing flows.
- The writing is very clear. All mathematical details are concise yet complete. The authors also provide enough context to distinguish their contribution from related works.
- The proposed method for dequantization and disjoint mixture modeling significantly improves performance with little additional computational cost.

Weaknesses:
There are no major flaws that I can think of. The paper can be further improved if the authors can provide a few alternatives to the current invertible mapping to convex polytopes and compare performance and computation costs. Other design choices can be explored as mentioned in the conclusion section, but the current content seems sufficient for a paper.

---

> ### Author Response · Authors · 2022-07-28
> **Response to Reviewer xCLG**
>
> We thank the reviewer for their encouraging comments! We do have other mappings in mind, but they all require much more computation than the simple one presented in the paper, so we decided not to pursue them for the current paper. In regards to other methods, it depends. It is difficult to beat the performance of large autoregressive models on discrete data, especially since they are universal models on discrete data. However, we are excited to see how this approach can be further improved for the disjoint mixture modeling direction, and can envision that it may become a cost-effective approach with tractability guarantees.

---

### Official Review · Reviewer_cFmy · 2022-07-11

**Rating:** 5
**Confidence:** 4
**Soundness:** 3 good
**Presentation:** 2 fair
**Contribution:** 2 fair

**Summary:**

The paper proposes a new method for modelling discrete data using normalizing flows by extending previous works on dequantization based method (like argmax flows) using a "tessellation" (not introduced or defined in the paper) based approach that automatically learns the boundaries for quantization. This is explore both for discrete to continuous and continuous to discrete.

**Questions:**

Please see in the strengths and weakness section.

**Limitations:**

Limitations have been discussed.

**Strengths And Weaknesses:**

**Strengths:**
- The paper describes sound extensions to normalizing flows for modelling discrete data and explores novel ways of incorporating tessellation based approaches in this context.
- The experiments are done on a variety of datasets including language modelling etc to study the empirical performance of the method.
- Significance: Getting good performance of generative models on discrete data is challenging and thus the problem is quite hard. This paper explores an interesting extension to previous methods to tackle this problem.

**Weakness:**
- Unfortunately, the paper is not clear on technical details. Important concepts are assumed to be known by the reader instead of properly introducing them e.g. tessellation, anchor points, Voronoi tessellation etc. The paper will greatly benefit from having a brief notations and definitions section. I also believe the paper will benefit greatly from having intuitive explanations accompany technical details to make it easier to follow.

- Similarly, several appropriate references are missing e.g. for dequantization, the paper by Uria, 2013 needs to be cited. Similarly, statements/claims as in line 46-48 need to be supported with relevant references.

- It was not immediately clear to me why is it useful to go from continuous case to the discrete case? What applications will this be important in? Apart from testing this on simple UCI datasets, I do not see any specific experiments for this.

- I also found the experiments fairly limited in scope. There are no experiments to show the capability of the model on modeling categorical data with large number of classes. The language modeling task have small vocabularies. I'd recommended to evaluate the model on word-based language models with large vocabularies.

- Similar to RAD flows, I believe the method will suffer from exploding gradients near the boundaries. How is the training stabalised to address this behaviour?

---

> ### Author Response · Authors · 2022-07-28
> **Response to Reviewer cFmy**
>
> We thank the reviewer for their candid comments, especially for their concrete suggestions for improving presentation. We have improved the paper in light of these comments.
>
> > Unfortunately, the paper is not clear on technical details. Important concepts are assumed to be known by the reader instead of properly introducing them e.g. tessellation, anchor points, Voronoi tessellation etc. The paper will greatly benefit from having a brief notations and definitions section. I also believe the paper will benefit greatly from having intuitive explanations accompany technical details to make it easier to follow.
>
> We defined anchor points and Voronoi tessellation in Eq 4. However, we agree we did not define tessellation succinctly, and appreciate the reviewer bringing this up. We have now included an English description in the introduction, right after the first mention of tessellation, as well as a more technical definition of Voronoi tessellation after Eq 4. If these definitions are unsatisfactory, we can make further adjustments.
>
> Furthermore, we’ve added an example illustration (Fig 2) to accompany technical details when discussing the invertible mapping. We hope this will make the paper easier to follow.
>
> > Similarly, several appropriate references are missing e.g. for dequantization, the paper by Uria, 2013 needs to be cited.
>
> If the reviewer believes that more appropriate papers should be referenced, we can certainly update our paper.
>
> We’ve added the reference to Uria 2013; however, this paper only adds uniform noise without discussing the implications. Our choice of citation on dequantization (Theis 2016) is mainly due to their specific discussion on dequantization and the relation between the original discrete distribution and the dequantized continuous density (as an ELBO), though it is easy to see the connection in hindsight.
>
> > Similarly, statements/claims as in line 46-48 need to be supported with relevant references.
>
> It is difficult to find a citation to support a statement about how something does not exist. However, we agree that such a statement may be too strong. We’ve added some clarifications and weakened this statement.
>
> > It was not immediately clear to me why is it useful to go from continuous case to the discrete case? What applications will this be important in? Apart from testing this on simple UCI datasets, I do not see any specific experiments for this.
>
> The main argument for disjoint mixture models is the ability to scale without extra computation costs. Our models on the UCI data set are effectively disjoint mixtures of normalizing flows. This approach is simple and does not increase compute cost by much. Whereas a naive method of constructing a mixture of normalizing flows would quickly become intractable as it requires a much larger compute/memory cost (costs increase linearly with respect to the number of components).
>
> Granted, the UCI experiments are not very exciting; we agree. We tried scaling up to higher dimensions, and building deep disjoint mixture models, but found that the choice of Voronoi tessellation did not lead to better performance. Our hypotheses for the reasons behind this are discussed in the limitations section, and we plan on improving this general direction in the future.
>
> > I also found the experiments fairly limited in scope. There are no experiments to show the capability of the model on modeling categorical data with large number of classes. The language modeling task have small vocabularies. I'd recommended to evaluate the model on word-based language models with large vocabularies.
>
> The itemset modeling experiments contain vocabulary sizes of 200-800, and are used to test the ability to model larger vocabularies. While the baseline methods with fixed dequantization require binarizing the data beforehand—this is a manual way of working with a large vocabulary since otherwise the embedding space would be just as high—our method with learnable dequantization boundaries works automatically without needing this binarization trick, which can create many empty classes if the original number of the classes is not a power of 2.
>
> We mainly used language modeling as an experiment since it has been used by prior works and has reproducible code. We showed that by simply changing the dequantization method we obtain meaningful gains over the prior work. This result is shown for a variety of data modalities.
>
> > Similar to RAD flows, I believe the method will suffer from exploding gradients near the boundaries. How is the training stabalised to address this behaviour?
>
> The gradients theoretically do explode, but we did not find this to be a huge problem in practice. We believe this is because the density itself is still continuous and differentiable almost everywhere, and that most samples will appear in the interior rather than on the boundary. We also do gradient clipping, so in the rare cases where a sample is extremely close to the boundary, it would get clipped.

---

> > ### Comment · Reviewer_cFmy · 2022-08-07
> > **Author's response**
> >
> > Thank you to the authors for the responses. I still think the manuscript will benefit from more thorough empirical analysis. Having said that, I believe it is still a nice work. I am increasing my score to 5.

---

### Official Review · Reviewer_vHMS · 2022-07-11

**Rating:** 8
**Confidence:** 3
**Soundness:** 3 good
**Presentation:** 4 excellent
**Contribution:** 4 excellent

**Summary:**

A dequantization method is presented that is able to map between discrete and continuous distributions using continuous, invertible normalizing flows. Two applications are presented, namely Voronoi dequantization and Voronoi mixture models.

**Questions:**

In Figure 5, the caption could directly explain what "Samples" refers to. Line 325 also mentions how the method could potentially make gradient-based optimization difficult, has this been tried in any experiments? It would be interesting to see such results, or these kinds of applications with the framework.

**Limitations:**

The limitations are clear.

**Strengths And Weaknesses:**

The results are well founded, and figures visually explain the concepts well. Though the text is often a bit convoluted, where the text explanation tends to be over-complex and hard to understand for non-experts in the field.

Although it is great how much space is given to explaining the background and catching the reader up, more results and experiments would perhaps be even more intriguing.

Could it be that no Appendix was present with the submission, even though it was referred to?

---

> ### Author Response · Authors · 2022-07-28
> **Response to Reviewer vHMS**
>
> We thank the reviewer for their encouraging comments! We agree the paper is rather dense due to the amount of information and number of equations. We’ve updated the paper to include a new illustration of the technical method (Figure 2), and also simplified some of the exposition in the paper.
>
> > Could it be that no Appendix was present with the submission, even though it was referred to?
>
> The Appendix is in the supplementary materials, as we are not sure if NeurIPS allows appendices to be included in the main PDF. Mainly proofs and experimental details are provided there.
>
> > In Figure 5, the caption could directly explain what "Samples" refers to.
>
> These are samples obtained from the model. Mainly shown to verify correctness of the implementation, that the density evaluation correctly corresponds to the sampling process of the model.
>
> > Line 325 also mentions how the method could potentially make gradient-based optimization difficult, has this been tried in any experiments? It would be interesting to see such results, or these kinds of applications with the framework.
>
> These discontinuous gradients were a concern a priori and during experimentation; however, we never really encountered any problems. To clarify, we believe that while the density (i.e. training loss) of the model is continuous, its gradient is not. But this is a fairly common phenomenon within deep learning, e.g. when using ReLU neural networks. Nevertheless, since adaptive optimizers estimate second-order information, it’d be interesting to see if there are any meaningful improvements by making the density of the model smoother in future works.

---

> > ### Comment · Reviewer_vHMS · 2022-08-09
> > **Response to Authors**
> >
> > Thank you for the revision! The newly added figure 2 does explain the concept in much simpler terms. My apologies for not having found the supplementary material before, thank you for pointing it out. Including all these details in the appendix is indeed much appreciated!

---

### Official Review · Reviewer_qSwN · 2022-07-14

**Rating:** 7
**Confidence:** 3
**Soundness:** 4 excellent
**Presentation:** 4 excellent
**Contribution:** 3 good

**Summary:**

The paper presents a tessellation based approach to mapping between continuous and discrete distributions which learns quantization boundaries in continuous space. The approach has several desirable mathematical and computational properties that give rise to more efficient scaling in dimensions than previous approaches. The approach is used in two ways--Voronoi dequantization and Voronoi mixtures--which are demonstrated on data and shown to be competitive with baselines. The authors concluded by noting some limitations and directions for extension.

**Questions:**

Following the above discussion, I would be interested in understanding the advantage of the Voronoi approaches.

**Limitations:**

The authors do mention a couple limitations and directions for future work, which I applaud. There is no reason to discuss social impact for this work.

**Strengths And Weaknesses:**

The paper is a mostly clear and interesting presentation of an interesting approach to bridging discrete and continuous distributions. Strengths include: clear mathematical and general exposition, an interesting mix of mathematical and computational considerations, and interesting empirical tests. There are not many weaknesses, though I found myself wanting to understand the advantages of the Voronoi approaches better. The experiments did what the authors proposed (show competitiveness), but it would have been nice to have stronger intuition.

---

> ### Author Response · Authors · 2022-07-28
> **Response to Reviewer qSwN**
>
> We thank the reviewer for their encouraging comments! We started exploring this tessellation approach for its ability to learn the boundaries themselves, whereas---to the best of our knowledge---prior dequantization approaches all had manually designed boundaries. In the future, we’re excited about using tessellation to build scalable disjoint mixture models. Most generative models do not tend to perform well on multimodal distributions or distributions with holes as the mapping from noise to data is necessarily continuous, while combining them via mixtures naively increases computational cost. At this time, we do not yet have a tractable method of building *deep* disjoint mixture models, but we believe a tessellation-based approach (not necessarily Voronoi however) can lead us there. In regards to building more intuition, we've added an illustration (Fig 2) to help readers better understand the method itself.

---

> > ### Comment · Reviewer_qSwN · 2022-08-04
> > **Response to authors**
> >
> > Thanks for your response!

---

### Meta-Review · Area_Chair_1TaD · 2022-08-25

**Recommendation:** Accept
**Confidence:** Certain

**Metareview:**

The authors develop a tesselation based approach to map between discrete and continuous spaces. They use this approach to dequantize data to port likelihood based models on continuous spaces to discrete spaces and to scale mixture models where each mixture component has disjoint support. From the view of normalizing flows, the approach is neat, and the results are supportive. The two outstanding things, I'd encourage to authors to work on are 1) the accessibility of the writing and 2) placing the work in the context of generative models outside of normalizing flows

**Award:**

No

---

### Decision · Program_Chairs · 2022-09-14

Accept